# Identification of Potentially Pathogenic Variants Associated with Recurrence in Medication-Related Osteonecrosis of the Jaw (MRONJ) Patients Using Whole-Exome Sequencing

**DOI:** 10.3390/jcm11082145

**Published:** 2022-04-12

**Authors:** Songmi Kim, Seyoung Mun, Wonseok Shin, Kyudong Han, Moon-Young Kim

**Affiliations:** 1Center for Bio Medical Engineering Core Facility, Dankook University, Cheonan 31116, Koreamunseyoung@gmail.com (S.M.); 2Department of Microbiology, Dankook University, Cheonan 31116, Korea; 3NGS Clinical Laboratory, Dankook University Hospital, Cheonan 31116, Korea; w.shin86@gmail.com; 4Department of Oral and Maxillofacial Surgery, College of Dentistry, Dankook University, Cheonan 31116, Korea

**Keywords:** bisphosphonates, medication-related osteonecrosis of the jaws, osteonecrosis of the jaw, single nucleotide polymorphism, whole-exome sequencing

## Abstract

Background: Bisphosphonates are antiresorptive and antiangiogenic drugs that prevent and treat bone loss and mineralization in women with postmenopausal osteoporosis and cancer patients. Medication-related osteonecrosis of the jaw (MRONJ) is commonly caused by tooth extraction and dental trauma. Although genetic and pathological studies about MRONJ have been conducted, the pathogenesis of MRONJ still remains unclear. Methods: We aimed to identify genetic variants associated with MRONJ, using whole-exome sequencing (WES). Ten MRONJ patients prescribed bisphosphonates were recruited for WES, and jawbone tissue and blood samples were collected from the patients. Results: The analysis of the WES data found a total of 1866 SNP and 40 InDel variants which are specific to MRONJ. The functional classification assay using Gene Ontology and pathway analysis discovered that genes bearing the MRONJ variants are significantly enriched for keratinization and calcium ion transport. Some of the variants are potential pathogenic variants (24 missense mutations and seven frameshift mutations) with MAF < 0.01. Conclusions: The variants are located in eight different genes (*KRT18*, *MUC5AC*, *NBPF9*, *PABPC3*, *MST1L*, *ASPN*, *ATN1*, and *SLAIN1*). Nine deleterious SNPs significantly associated with MRONJ were found in the *KRT18* and *PABPC3* genes. It suggests that *KRT18* and *PABPC3* could be MRONJ-related key genes.

## 1. Introduction

Medication-related osteonecrosis of the jaw (MRONJ) refers to an uncommon medical condition that can occur in patients receiving antiresorptive or antiangiogenic therapy to treat osteoporosis and diverse malignant conditions such as multiple myeloma, breast cancer, and prostate cancer [1]. When the first case of MRONJ was documented in 2003, only bisphosphonates were associated with the pathophysiology, and thus the condition was widely known as bisphosphonate-related osteonecrosis of the jaw (BRONJ) [2]. However, in light of the numerous recent reports of osteonecrosis cases associated with the use of various other antiresorptive and antiangiogenic agents such as denosumab, the current universal nomenclature for the disease has been modified to MRONJ. The antiangiogenic agents or targets of the VEGF pathway, such as the anti-VEGF monoclonal antibodies (Bevacizumab) and tyrosine kinase inhibitors (Sunitinib, Sorafenib), that are prescribed for the treatment of cancer-related conditions or organ rejection in renal transplant patients (Sirolimus) have also been indicated in the pathophysiology of the disease [3]. The key hypotheses of disease specificity for the jaws include bone remodeling inhibition, angiogenesis inhibition, inflammation, innate or acquired immune dysfunction, soft tissue toxicity, and genetic predisposition. Several hypotheses can explain the overall pathophysiology of MRONJ [4,5]. Antiresorptive and antiangiogenic reagents, including bisphosphonates and denosumab, inhibit bone remodeling by affecting osteoclast formation and differentiation, or they directly inhibit angiogenesis by reducing blood flow to the bone [5,6]. Genetic factors such as single nucleotide polymorphisms (SNPs) and mutations are significantly associated with MRONJ development by regulating bone remodeling, angiogenesis, and immune responses [7,8]. Prior attempts to determine the efficacy of a drug holiday in preventing MRONJ have yielded inconclusive results, and thus the evidence to support or refute the use of such a holiday remains limited [7]. The hallmark clinical feature of MRONJ is the presence of exposed necrotic bone in the oral cavity that persists for more than eight weeks without any signs of healing. This is frequently accompanied by concurrent infection signs such as pain, swelling, fistula formation, and in its severest form—pathological fractures (Figure 1). In some MRONJ patients, a specific symptom, known as the ‘numb chin syndrome’ may also manifest [8]. According to the current AAOMS guidelines, MRONJ can be classified into four distinct stages (stages 0 to III), for which the suggested methods of treatment differ. For early stages (stage I) of MRONJ patients, conservative treatment, including antibiotic and septic administration, and an antibacterial oral rinse, results in pain control and prevention of necrosis. Surgery may not be needed in the absence of disease progression, and if the patient has an adequate quality of life [4]. For patients suffering from advanced stage (stage II–III) MRONJ, invasive surgery, including jaw resection, can yield optimal mucosal healing; however, up to 25% of patients can suffer from severe postoperative morbidity following this type of extensive surgery. Therefore, to minimize morbidity by establishing novel prevention or early detection methods, recent studies have focused on determining the pathophysiology of MRONJ.

The incidence of MRONJ is relatively lower in the oral bisphosphonate group for osteoporosis than in the intravenous bisphosphonate group for cancer; hence, to date, many hypotheses concerning the pathogenesis of MRONJ have been raised by many authors. Previous studies reported that genetic polymorphisms in several candidate genes (*TGFb1*, *MMP2*, *CYP2C8*, *VEGF*, *COL1A1*, *RANK*, *OPG*, *OPN*, and *PPARG*) involved in the activation and differentiation of osteoclast and bone remodeling are significantly associated with MRONJ [1,9,10,11]. In addition, many researchers recently found that MRONJ is caused by multiple biological pathways involving immunoglobulin, cell adhesion, and cytoskeleton [12]. However, despite the numerous genetic studies, such as the genome-wide association studies (GWAS), whole-exome studies (WES), and the candidate gene studies (CGS), that have been performed to ascertain the condition’s exact pathogenic mechanism, the pathogenesis remains yet to be determined. Furthermore, although potential associations between the development of MRONJ and several genes have been documented in the previous literature, the GWAS, WES, and CGS have all failed to elucidate a single gene as the sole risk factor for MRONJ development.

Our study aimed to identify genetic variants associated with MRONJ by analyzing somatic and germline mutations through WES. By conducting an integrated analysis with the preceding research results, eight genes were determined to be closely related to MRONJ, and mutations in the genes were investigated in the present study. The result found that twelve and ten genetic mutations of *KRT18* and *PABPC3* genes, respectively, are predicted to modify the protein structure of the genes in MRONJ patients.

## 2. Materials and Methods

### 2.1. Ethics Statement

Experiments and analyses involving human subjects or human data in this study were approved by the Institutional Review Board of Dankook University Hospital (IRB No. 2018-01-009). All clinical investigations were performed in accordance with the Declaration of Helsinki. Written informed consent was obtained from all patients included in this study.

### 2.2. Patient Selections

Ten MRONJ patients (stage II–IIII) from the oral and maxillofacial surgery department of Dankook University Hospital participated in this study. The patients visited the Dankook University Hospital for surgical intervention from March 2018 to December 2018 and received bisphosphonates (BPs) therapy, either risedronate (9 cases) or alendronate (1 case). Patients were considered to have MRONJ if all of the following three characteristics were present according to the position paper issued by the American Association of Oral and Maxillofacial Surgeons (AAOMS) in 2014; (i) current, previous treatment with an antiresorptive or antiangiogenic therapy, (ii) non-healing, exposed bone in the maxillofacial region that had persisted for more than eight weeks, and (iii) no history of radiation therapy to the jaw. We also analyzed the characteristics of the clinical findings (medications, smoking, sex, stage) in a panoramic radiographic study (Figure 1).

### 2.3. DNA Preparation

Peripheral venous blood (2 mL) and routine jawbone specimens were obtained from the ten MRONJ patients in the Department of Oral and Maxillofacial Surgery of Dankook University Hospital. Genomic DNA was extracted from the blood and the lesion jawbone tissues immediately after surgical treatment of ten MRONJ patients using a DNeasy Blood & Tissue Kit (Qiagen, Germany) in accordance with the manufacturer’s protocols, and stored at −80 °C until use.

### 2.4. Whole-Exome Sequencing (WES) and Individual Variant Calling

For NGS library construction, each genomic DNA (1 μg) was sheared to 180–280 bp fragments using a DNA shearing system (Covaris, Woburn, MA, USA). Sequencing libraries were generated using a SureSelect Human All Exon Kit (Agilent Technologies, Santa Clara, CA, USA) following the manufacturer’s instructions. DNA that was ligated with adapters were enriched during PCR reaction. The WES libraries were captured and purified using streptavidin-coated magnetic beads and the AMPure XP system (Beckman Coulter, Beverly, MA, USA). WES was performed on the Illumina NovaSeq 6000 with a 2 × 150 bp paired end (PE) read.

Sequencing reads were mapped to the human reference genome (GRCh37/hg19) using the Burrows–Wheeler Alignment tool (BWA 0.7.8) with default parameters [13]. PCR duplicates were removed using Picard (http://picard.sourceforge.net/index.shtml, accessed on 8 November 2019). Variant calling was performed using the Genome Analysis Tool Kit (GATK v3.8) [14]. The functional annotation of variants was conducted using ANNOVAR (2015Dec14) [15]. To obtain genetic variants highly related to MRONJ, we compared two datasets with the variant candidates (Korean variant archive; KOVA and GSK project). KOVA was used as the control dataset to exclude the common Korean variants. The GSK project investigating the MRONJ-related variants found in the saliva of MRONJ patients was used to collect variants shared between MRONJ diagnosed patients.

### 2.5. Functional Gene Classification and Pathway Analysis

To investigate the biological relevance of the candidate genes, we performed gene ontology (GO) enrichment analysis to classify them into three classes, including biological process (BP), cellular component (CC), and molecular function (MF), by using Metascape software (https://metascape.org/gp/index.html, accessed on 19 February 2020) [16]. Then, the enrichment of candidate genes was conducted to identify pathways using the Kyoto Encyclopedia of Genes and Genomes (KEGG) database (http://www.genom e.jp/kegg/pathw ay.html, accessed on 19 February 2020). The significant function of genes was filtered by statistical cut-off using the adjusted *p*-value (FDR; False Discovery Rate).

### 2.6. Pathogenicity of Variants 

To investigate the occurrence and the frequency of the variants in the general population, three public databases were used as follows: ExAc (*n* = 60,706, http://exac.broadinstitute.org/, accessed on 14 April 2021), 1000 Genomes project phase 3 database (1000 G; *n* = 2504, http://www.internationalgenome.org/, accessed on 14 April 2021), and NHLBI Exome Sequencing Project (*n* = 6503, http://evs.gs.washington.edu/EVS/, accessed on 14 April 2021). Using the comparison of the three control databases, we investigated the frequency of the genetic variations identified in MRONJ patients and selected unique variants with minor allele frequencies (MAF) < 0.01. In addition, the pathogenicity of the variants was analyzed to predict functional effects sorting intolerant from tolerant (SIFT) (http://sift.jcvi.org, accessed on 14 April 2021), using Polymorphism Phenotyping v2 (PolyPhen-2) (http://genetics.bwh.harvard.edu/pph2/, accessed on 14 April 2021), a Mutation Assessor, and a Mutation Taster. 

Furthermore, we predicted protein structures using the AlphaFold protein structure database (https://alphafold.ebi.ac.uk/, accessed on 10 August 2021), Phyre2 (http://www.sbg.bio.ic.ac.uk/phyre2/html/page.cgi?id=index, accessed on 10 August 2021). In addition, we used UCSF chimera (https://www.cgl.ucsf.edu/chimera/, accessed on 10 August 2021) and DynaMut2 (http://biosig.unimelb.edu.au/dynamut2/, accessed on 10 August 2021) to analyze the protein stability and protein conformational changes [17,18]. The filtration and prioritization framework used for data analysis is described in Appendix A.

## 3. Results

### 3.1. Clinical Findings

The clinical characteristics of the ten MRONJ patients are shown in Table 1. One man and nine women were enrolled in this study, and none of them were smokers. The mean age was 77.8 ± 5.64 years. Of the ten MRONJ patients, the mandible bone was involved in seven (70%), and the rest were maxillary bones. One of the ten MRONJ patients was spontaneous, eight had teeth extractions, and the other had root canal therapy prior to the onset of MRONJ. Only one patient was taking alendronate, and the others were taking risedronate (Table 1). Ten patients developed the disease after long-term administration for at least four years.

### 3.2. Whole-Exome Sequencing and Quality Controls

WES data were generated with an average 11.4 Gb of sequences per sample on the Illumina NovaSeq6000 platform. Sequencing quality for Q20 and Q30 was an average of 96.75% and 91.99%, respectively (Appendix A). Of the WES reads, 97.34% were mapped to the human reference genome. The blood and lesion samples (i.e., the lesion jawbone tissues) were sequenced to an average of 104.16× depth on target, and an average of 99.57% of exonic regions were covered with at least 10× depth on target (Appendix A).

### 3.3. Identification of Single Nucleotide Polymorphism (SNPs) Related to MRONJ

From WES data, an average of 433,602 germline SNP variants and 367,806 somatic SNP variants were identified in ten patients (Appendix A). To identify variants associated with MRONJ, we used a strict filtering method that collects common variants from all ten patients. We identified a total of 39,720 germline SNP variants and 32,225 somatic SNP variants shared by all ten patients. Then, to rule out the common variants in Korean, we compared the total variants with the KOVA dataset containing 293,049 variants in 1055 healthy Korean individuals [19]. The 219,772 MRONJ-associated variants discovered from the GSK project with 142 saliva samples of MRONJ patients [20] were used for variant recalibration and separation of somatic mutations from germline mutations (Figure 2). Through the analysis, we identified a total of 4171 SNPs in 2321 genes associated with MRONJ, including 3876 SNPs detected in all three datasets (saliva, blood, and lesion), 216 SNPs detected in both blood and saliva datasets, and 79 SNPs detected in both saliva and lesion datasets. Additionally, we collected 1151 lesion-specific SNPs to identify somatic variants (Table 2, Figure 2). We found 1020 nonsynonymous SNP variants, including the missense (*n* = 946), nonsense (*n* = 8), and unknown mutations (*n* = 66) in the exonic and splicing region (Table 3, Appendix A).

In the MRONJ common group, we found 942 nonsynonymous variants, including missense, nonsense, and unknown variants, and ten frameshift InDels in the MRONJ common group (saliva, blood, and lesion). The top ten genes that accumulated a large number of nonsynonymous variants were *PDE4DIP*, *HLA-C*, *HLA-B*, *GPRIN2*, *KCNJ12*, *KCNJ18*, *HLA-A*, and *MST1L* genes (Table 3, Appendix A). Twenty-three of 942 nonsynonymous variants were distributed in *PDE4DIP*, which encodes anchoring the primary c-AMP-hydrolyzing enzyme and the phosphodiesterase 4D (PDE4D) protein, to the Gogi/centrosome region of the cell. Nine nonsynonymous variants were identified in *KCNJ18* or *KCNJ12*, which regulate potassium concentration in the skeletal muscle. KCNJ12 is an inwardly rectifying K+ channel activated by phosphatidylinositol 4,5-bisphosphate that controls the concentration of calcium ions in the cytosol of skeletal muscle [21]. *KCNJ18* encodes the skeletal muscle inward-rectifier K+ channel Kir2.6, which is regulated by thyroid hormones (T3) [22,23]. Three variants (R39Q, R40H, I249V) in the *KCNJ18* gene were previously reported, and they are related to thyrotoxic periodic paralysis (TPP) [22,24]. 

We identified nine lesion-specific missense variants in eight genes, including *PRAMEF1* (c.386 C>T, p.T129M and c.412T>C, p.C138R), *IGFN1* (c. 5047G>A, p.G1683R), *OR2T12* (c. 205A>G, p.M69V), *PABPC3* (c.878T>G, p.V293G), *PLIN4* (c. 787G>A, p.G263S), *LONRF2* (c.330C>G, p.D110E), *MUC12* (c.2564G>A, p.R855H), *FOXD4* (c.614C>T, p.P205L) (Table 3; Appendix A). PRAMEF1, a member of *PRAME*, is related to reproductive tissues during development. PABPC3, a cytoplasmic PABP, is associated with mRNA translation and stability that participates in nonsense-mediated decay. *MUC12*, a member of the mucin family, encodes an integral membrane glycoprotein that contributes to host cell defense infected by the bacterium and is associated with epithelial cell protection and regulation of cell growth. Ladeberg et al. (2008) reported that bisphosphonate inhibits the proliferation of keratinocytes in the oral mucosa [25]. Although no significance has been identified in the pathogenic effects and the phenotypic difference of these genes in MRONJ, they may affect biological functions such as cell morphology, proliferation, and differentiation, broken by protein structural alteration in protein–protein interactions (PPIs).

### 3.4. Identification of Insertion/Deletions (InDels) Variants Related to MRONJ

As for InDel variants, we identified an average of 72,754 germline InDels and 72,715 somatic InDels from WES data of ten patients (Appendix A). A total of 4901 germline InDels and 3892 somatic InDels are common in all ten MRONJ patients (Figure 2). Finally, a total of 1641 InDels in 1142 genes were only discovered in MRONJ patients, except for common Korean variants as we described above, which include 265 InDels detected in all three datasets (saliva, blood, and lesion), 241 InDels detected in both saliva and lesion dataset, and 46 variants detected in both saliva and blood dataset. Additionally, we included 1089 lesion-specific InDels, which were somatic variants. Of these, we collected 132 exonic InDel variants to identify variants that affect MRONJ, including common InDels in the saliva and blood (*n* = 4), common InDels in the saliva and lesion (*n* = 42), common InDels in the saliva, blood, and lesion (*n* = 40), and lesion-specific InDels (*n* = 46). They contain 32 frameshift InDels, 53 non-frameshift InDels, 46 unknown mutations, and one nonsense mutation (Table 3, Appendix A).

In MRONJ common group, ten frameshift InDels are identified in nine genes, including *MST1L* (c.894_897del, p.G298fs and c.119_123del, p.Q40fs), *HRNR* (c.1delA, p.M1fs), *NUDT18* (c.111delG, p.R37fs), *GPATCH4* (c.1083_1084insGT, p.F362fs), *ATG3* (c.920dupT, p.L307fs), *CYFIP2* (c.280dupC, p.I93fs), *NOP16* (c.586_587insAC, p.R196fs), *SSPO* (c.9204dupC, p.S3068fs), and *OR2T35* (c.609_615del, p.C203fs) (Table 3). HRNR is a member of the S-100 fused protein family, involved in protein phosphorylation regulation, cytoskeleton component, calcium homeostasis, and cell proliferation [26]. ATG3, a ubiquitin-like-conjugating enzyme, is involved in autophagy-related to osteoblast differentiation and mineralization, which plays an important role in bone homeostasis [27]. The representative genes harbored severe frameshift InDels loading that can impair these roles in genes that are vital for bone formation and control.

Eleven lesion-specific frameshift InDels were also identified in eight genes, which were *SARM1* (c.549_550del, p.G183fs; c.544_545insGC, p.S182fs), *SLAIN1* (c.219_220insGG, p.A73fs; c.229dupC, p.Q76fs), *ZFPM* (c.1330_1331del, p.E444fs; c.1335delT, p.P445fs), *CTU2* (c.1097_1097del, p.R366fs), *GOLGA6L2* (c.2080delA, p.R694fs), *LOC283710* (c.75_76del, p.P25fs), *PRDM15* (c.262dupC, p.R88fs), and *SCARF2* (c.2247dupG, p.P750fs). *SARM1* is highly associated with the neuron degeneration that mediates neuronal cell death. A loss-of-function (LOF) mutation (C844Y) in *PRDM15*, which regulates NOTCH and WNT signaling pathways, was identified in patients with brain malformations [28]. *SCARF2*, the scavenger receptor type F family, containing epithermal growth factor (EGF)-like domains and putative N-glycosylation sites contributes to lipid metabolism and intracellular signaling pathways in pathophysiological conditions. Two genetic variants (c.773G>A in exon 4 and c.1328_1329delTG in exon 8) of *SCARF2*, influencing cytoskeletal organization, were identified in patients with Van Den Ende-Gupta syndrome [29,30]. Interestingly, *SCARF2* is expressed in the mandibular, maxillary, and urogenital ridge tissues in mice at ten embryonic days and human endothelial cells [30].

We also identified 33 lesion-specific non-frameshift InDels in 26 genes, which were *GOLGA6L2* (*n* = 3), *NCOR2* (*n* = 3), *ASPN* (*n* = 2), *ATN1* (*n* = 2), *MPRIP* (*n* = 2), *GOLGA8R* (*n* = 1), and *ZFPM1* (*n* = 1) (Table 3; Appendix A). NOCR2, a member of thyroid hormone and retinoic acid receptor-associated corepressors, regulates androgen receptor signaling and androgen-induced cell proliferation. *ASPN* is related to the regulation of periodontal ligament differentiation and mineralization that maintains homeostasis of the tooth-supporting system [31]. Asporin protein binds to collagen fiber, resulting in induction of mineralization and inhibition of TGF-b in the cartilage differentiation [32]. ATN1 is related to Dentatorubral pallidoluysian atrophy (DRPLA) disease, a rare neurodegenerative disorder, which occurs with the expansion from 7–35 copies to 49–93 copies of a trinucleotide repeat (AAG/CAA).

### 3.5. Enrichment and Pathway Analysis

To understand the functional relevance of the isolated genetic variants focusing on MRONJ, we performed Gene Ontology (GO) and Kegg pathway analyses on 665 genes containing 1020 nonsynonymous SNP variants. As shown in Figure 3, the most significant gene set enrichment in the biological processes (BP) was ‘detection of stimulus’, ‘keratinization’, and ‘calcium ion transport’. Interestingly, the ‘protein kinase C signaling pathway’ was also elected in the biological processes (Figure 3; Appendix A). As referred to in the previous study, the bisphosphonates inhibit protein kinase C signaling, resulting in the accumulation of calcium and degradation of osteoclast [33]. The most significant enriched function was ‘ECM structural constituent’ for the molecular function (MF), followed by ‘olfactory receptor activity’, and ‘calcium ion binding’ (Figure 3; Appendix A). For the cellular component (CC), the function of genes associated with ‘keratin filament’ was most significantly enriched (Figure 3; Appendix A). For the Kegg pathway, the top-ranked significant pathways were ‘olfactory transduction’, ‘autoimmune thyroid disease’, and ‘ECM-receptor interaction’ (Appendix A). One of the primary mechanisms, the extracellular matrix (ECM), comprises over 90% of type I collagen, the most abundant collagen, in human bone. It is a quite meaningful result because Type I collagen is synthesized by osteoblasts, which drives bone tissue remodeling during wound healing.

Furthermore, we examined the interaction between the classified functions of MRONJ-specific variants by function network analysis using ClueGO (*p*-value < 0.05; Benjamini-Hochberg). The integrative function network analysis showed that the eight parent functions, including ‘integral component of luminal side of endoplasmic reticulum membrane’, ‘O-glycan processing’, ‘sensory perception of the chemical stimulus’, ‘keratin filament’, ‘calcium ion transport’, ‘protein kinase A signaling’, ‘phenylpropanoid metabolic process’, and ‘post-synaptic specialization organization’ were mainly clustered with 665 genes (Appendix A).

### 3.6. Pathogenicity Analysis

For the private or rare variants (MAF < 0.01), we also utilized three databases, including the 1000 genomes project database, NHLBI exome sequencing project (ESP), and ExAC database. We filtered in the candidate variants with allele frequencies lower than 0.01 or no allele frequency in all three databases. A total of 24 candidate SNPs and seven candidate InDels were selected in four genes (*KRT18*, *MUC5AC*, *NBPF9*, and *PABPC3*) and five genes (*MST1L*, *ASPN*, *ATN1*, *PABPC3*, and *SLAIN1*), respectively (Table 4).

In addition, we analyzed 24 rare SNP variants for the annotation of potentially deleterious effects using SIFT and polyphen2. Of these, three SNP variants (p.G38C, p.G43R in *KRT18*, and p.K231E in *PABPC3*) were predicted to be ‘deleterious’ in the SIFT, and ‘possibly damaging or probably damaging’ in Polyphen-2 (Table 4). Six SNP variants (p.R27W, p.P28Q, p.A32S, p.S34R in *KRT18*, and p.A181T, p.L207F in *PABPC3*) were predicted to be either deleterious or benign in the SIFT and Polyphen-2 (Table 4). KRT18 encoding the type 1 intermediate filament chain regulates protein synthesis, organelle positioning, and protection from apoptosis and necrosis. PABPC3 is a member of RNA binding proteins involved in the cytoplasmic regulatory process of mRNA metabolisms, such as RNA stability and translation initiation. RBMS3 (the c-myc gene single-strand binding protein), a paralog for *PABPC3*, regulates myc/ras cooperative transforming activity, apoptosis, DNA replication, and cytoplasmic RNA metabolism. GWAS studies showed that two SNPs (rs17024608 and rs10510628) in the *RBMS3* were associated with a bisphosphonates-induced reduction in collagen formation, osteonecrosis disruption, and bone turnover [34,35].

To predict protein structural alteration by amino acid (AA) changes, protein conformational changes in WT and mutants of *KRT18* and *PABPC3* were predicted using the AlphaFold protein structure database, phyre2 web tool. Nine rare SNP variants (p.R27W, p.P28Q, p.A32S, p.S34R, p.G38C, and p.G43R in *KRT18*, and p.A181T, p.L207F, and p.K231E in *PABPC3*) were structurally changed (Figure 4A). In addition, we calculated the distance between the atoms of two residues to predict the effect of the mutations on the distances between each pair of residues using the UCSF chimera tool. The distances between wild-type and mutations were changed in *KRT18* and *PABPC3* (Figure 4B). Notably, we identified pairs of charged residues that were significantly skewed from 8.68 Å to 4.67 Å by the R27W mutation (Figure 4B). 

Furthermore, we used DynaMut2 to analyze the entropy energy change between wild-type and mutant structures (in kcal/mol/K). All six substitutions (p.R27W, p.P28Q, p.A32S, p.S34R, p.G38C, and p.G43R), located in the head domain, which is a primary phosphorylation site of *KRT18*, have weak effects (−0.5 < ΔΔG < 0.5 kcal/mol) on KRT18 stability (Figure 5). The phosphorylation of *KRT18* at serine 33 and 34 regulate filament stability, keratin reorganization, binding to 14-3-3 protein during mitosis, and keratin protein turnover during apoptosis [36,37,38]. The glycosylation at serine 30 and 31 protects epithelial tissue from injury [37]. The loss and mutations of *KRT18* contribute to the phenotypes of several human diseases, including skin disorders, chronic renal disease, and liver cirrhosis [38].

In addition, three substitutions (A181T, L207F, and K231E) are located in the RNA recognition motifs (RRMs) of *PABPC3* that bind the poly(A)tail of single-stranded RNAs (Figure 4A). Unfortunately, the entropy change of PABPC3 could not be analyzed because the intact protein structure of PABPC3 could not be obtained from the protein data bank (PDB). However, a p.K231E substitution located in the RNP1 motif ((K/R)-G-(F/Y)-(G/A)-F-V-X-(F/Y)), a highly conserved region, may interact with the translation initiation factor and regulatory proteins, affecting translation initiation and nonsense-mediated decay.

## 4. Discussion

Bisphosphonates, known as a primary drug for the onset of MRONJ, is the antiresorptive agent that prevents loss of bone density. They are used to treat osteoporosis, osteogenesis imperfecta, primary hyperparathyroidism, and cancer. The bisphosphonates bind preferentially to calcium ions in the bones and accumulate in high concentrations. However, the bisphosphonates cause mucosal ulceration and exposure to the necrotic bone, leading to osteonecrosis of the jaw. Previous studies reported that the bisphosphonates induce senescence and apoptosis of the cells by blocking the cholesterol biosynthetic pathways (mevalonate pathway) that indirectly inhibit the blood supply of the bone [20,39,40]. Bisphosphonates, which are closely related to anti-angiogenesis, affect the structure of soft tissues in the jaw, causing soft tissue cell damage. In this study, we identified the candidate genes and sorted out the variants closely associated with MRONJ in lesion and blood samples through WES. Although previous studies have constantly attempted to screen the candidate genes and variants that show a significant correlation with MRONJ using GWAS, WES, and CGS, the single genes and variants associated with MRONJ development have not been identified; therefore, the need for in-depth research on the onset, development, diagnosis, and treatment of MRONJ is drawing attention by securing various genetic data according to various races, drugs, living environments, and clinical aspects.

In function enrichment analysis for MRONJ-specific variants, the protein kinase C (PKC) signaling pathway, that can either read out lipid signals alone or combine the ability to read out simultaneous calcium ions, is highly enriched in the biological processes. The calcium ion is a second messenger that regulates differentiation and activation of osteoclasts according to calcium concentration [41]. The calcium ion signaling regulates the activation of PKC and NFATc1 in osteoclasts, which is involved in cell proliferation and differentiation [33]. Previous studies reported that treatments of bisphosphonates, such as alendronate and risedronate, were accumulated in the bone matrix. They trigger osteoclast apoptosis, resulting in cytoskeletal regulation [42,43]. Human parathyroid hormone (PTH) is highly associated with the regulation of calcium metabolism; thus, when calcium ions increase in the serum, the osteoclast activates, which degrades bone. These results suggest that the calcium ion is closely related to MRONJ development, regulating cell proliferation and differentiation. Through our study, we were able to observe variations in the following regulatory factors: FLT4, MC1R, ULK4, ADGRV1, TTN, MYOM1, and AIP, closely related to PKC, which respond immediately to calcium signals in osteoclast activity.

Through the systematic pathogenicity analysis, we identified 24 candidate SNPs and seven candidate InDels in four genes (*KRT18*, *MUC5AC*, *NBPF9*, and *PABPC3*) and five genes (*MSTIL*, *ASPN*, *ATN1*, *PABPC3*, and *SLAIN1*), respectively. KRTs are intermediate filament proteins found in skin and epithelial tissues that play an important role in constructing the cytoskeleton of the cells. KRTs maintain the structure of skin cells and other epithelial tissues by regulating electrolyte transport, post-translational modification, and prevention of degradations; however, disruption of KRTs leads to skin and mucosal diseases. As shown in the results of previous studies, the genetic effect of *KRT4* and *KRT13* mutations, a member of the keratin family, is mainly implicated in the pathogenesis of white sponge nevus (WSN) disorder, a rare autosomal dominant disorder in oral mucosa [44,45]. 

*KRT18* is also involved in the cytoskeletal signaling pathway, estrogen signaling pathway, and staphylococcus aureus infection pathway [46,47,48,49]. Estrogen generated from the mevalonate pathway contributes to lipid metabolism, producing cholesterol, fatty acids, and steroid hormones, inducing osteoblast proliferation, osteoclast apoptosis, and stimulating matrix synthesis associated with MRONJ [50,51]. Bisphosphonates could influence the production of estrogen, a type of steroid hormone, by inhibiting the mevalonate pathway, and reduced estrogen exacerbates age-related bone loss in women [52]. In addition, a previous study reported that the microbial films of predominant bacteria, including *Fusobacterium*, *Actinomyces*, *Staphylococcus*, *Streptococcus*, *Selenomonas*, and *Trepenemes*, were present in osteonecrosis of MRONJ patients [53,54,55,56]. Disruption of *KRT18* could be made to fail to interact with CDH1 (calcium-dependent cell adhesion proteins) and EPCAM (epithelial cell adhesion molecule), which are involved in the proliferation of epithelial cells and defense against mucosal infection, respectively, which affects filament reorganization; therefore, disruption of *KRT18* may influence the cytoskeleton of the skin and mucosal, leading to oral mucosa diseases, including MRONJ. 

Here, we accomplished the first rare-variant association study for MRONJ in lesions isolated from human jawbone tissue and blood samples. Our genetic expectation demonstrated that the *KRT18* and *PABPC3* could be potential candidate genes related to MRONJ and developing the MRONJ phenotype. This information could help clinicians to exclude patients who may develop MRONJ during dental procedures and take special preventive measures. Furthermore, it would be an innovative action to prevent drug side effects caused by genotypes through drug sensitivity diagnosis in patients, before the prescription of osteoporosis treatment drugs. 

As a limitation of the current research, first, the number of samples in this study was not large enough to detect statistically significant genetic variants. If more robust data is accumulated from patient cohorts with similar clinical environments and conditions, it is believed that it would provide a strong basis from which to derive genetic implications. Second, examining the functional association with germline and somatic variants focusing on MRONJ might provide a crucial message to identify alterations that can be issued in the transcription or translation stage.

## 5. Conclusions

We performed whole-exome sequencing to identify MRONJ-associated variants. We identified 954 nonsynonymous SNP variants and 40 InDels accompanied with 691 genes, shared by MRONJ patients in this study. Of these, we found significant genetic variations between MRONJ patients and control sets, and we examined their function through analysis of GO classification and KEGG pathways. The genes bearing the variants were highly related to biological functions of the cytoskeleton, calcium-ion binding, and keratinization. Notably, nine variants located in *KRT18* and *PABPC3* genes were considered pathogenic variations, influencing immoral protein changes. Our finding suggests that *KRT18* and *PABPC3* may be implicated in MRONJ pathophysiology, and that they may be used as a therapeutic target in the treatment of osteonecrosis. It is clear that further studies using more cases and controls are needed to confirm the molecular biological association between *KRT18* and *PABPC3* in the development and occurrence of MRONJ, but we have demonstrated new proven genetic changes to understand the pathogenesis of MRONJ. 

## Figures and Tables

**Figure 1 jcm-11-02145-f001:**
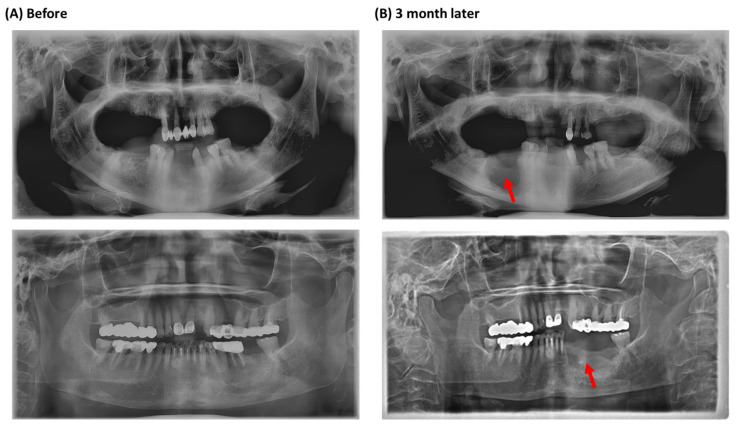
Panoramic radiographic images of MRONJ patients in this study. (**A**) Extraction of the mandibular molars. (**B**) Three months after the extraction. Red arrows point to the necrotic area.

**Figure 2 jcm-11-02145-f002:**
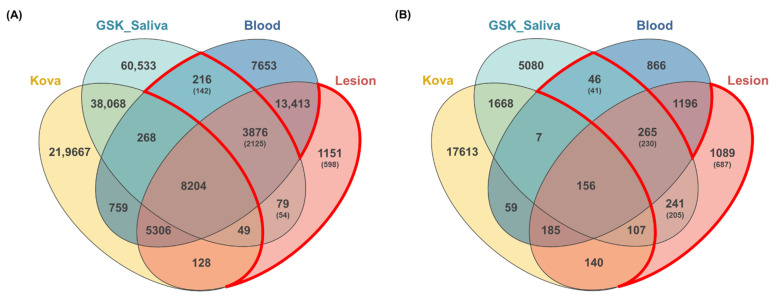
Comparisons of the variants among four datasets. The number of variants identified from bidirectional approaches (blood and lesion tissue) compared with the previous data set from the KOVA and GSK-saliva. Overlapping areas in the Venn diagram represent common variants between the comparison groups. (**A**) SNPs variants; (**B**) InDels variants.

**Figure 3 jcm-11-02145-f003:**
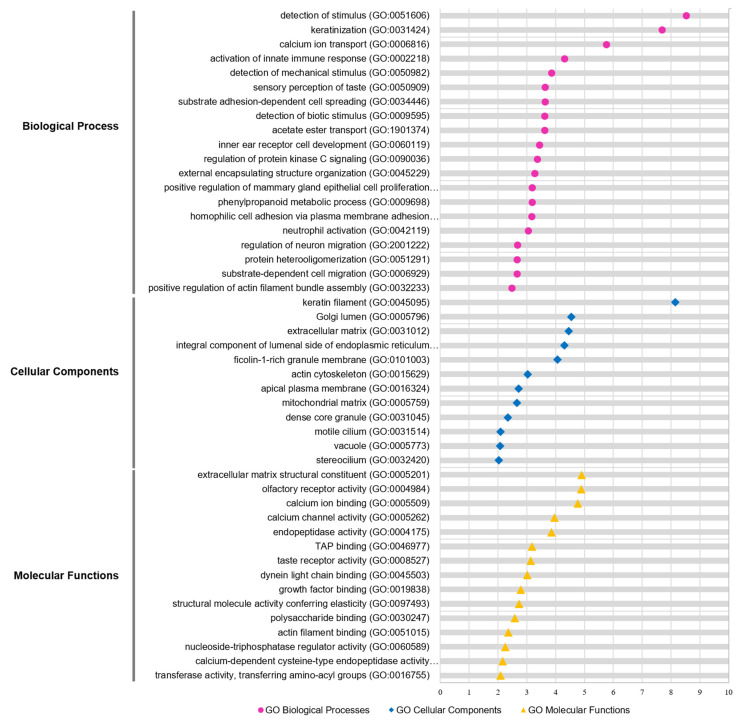
GO functional analysis of candidate genes. GO enrichment analysis of WES genes was retrieved using Metascape software. Significantly (*p* < 0.05) enriched GO terms involved in biological process (pink), cellular component (blue), and molecular function (yellow) and branches are presented. The adjusted statistically significant values were negative 10-base log-transformed. The X-axis represents the enrichment scores of the terms −log10(*p*-value), and the Y-axis represents the enriched GO terms in the pathway. P-value was adjusted by Benjamini–Hochberg FDR. GO, gene ontology.

**Figure 4 jcm-11-02145-f004:**
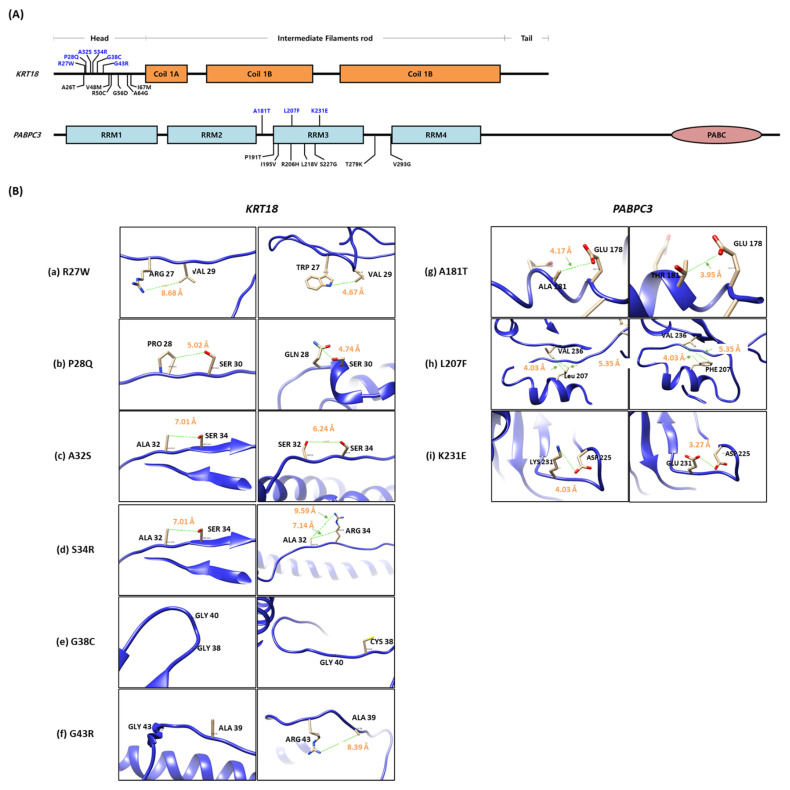
Comparison of the wild-type protein structure with the mutant. (**A**) Gene structures. Blue characters indicate rare variants that have MAF < 0.05. (**B**) Structural comparison of the distance between wild-type (left side) and mutants (right side) in *KRT18* (**a**–**f**) and *PABPC3* (**g**–**i**) protein. The adjacent amino acid is noted by black words, and the predicted distance between two residues in the protein structure is noted by orange words.

**Figure 5 jcm-11-02145-f005:**
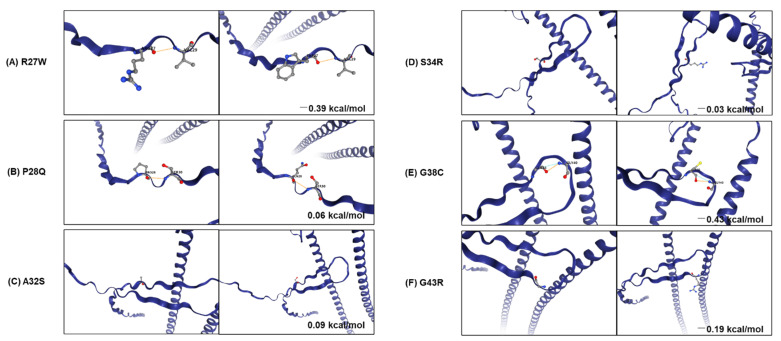
Analysis of stability and dynamic changes between WT and mutants in *KRT18*. Stability and dynamic changes between WT (**A**–**C**) and five mutations (**D**–**F**) of *KRT18* were analyzed. The figure shows the change in Gibbs free energy (in Kcal/mol) for mutations and non-covalent interactions established by the mutated residue. The orange and blue dash lines indicate polarity and van der Waals interactions, respectively.

**Table 1 jcm-11-02145-t001:** Clinical characteristics of patients with MRONJ.

Patient No.	Gender	Age	BPs Administered	Duration of Administration (Years)	Affected Area	Type of Dental Intervention	Stage	Smoking
1	Male	70	Risedronate	10	Maxilla and Mandible	Tooth extraction	III	No
2	Female	77	Risedronate	10	Mandible	Tooth extraction	II	No
3	Female	86	Risedronate	15	Mandible	Tooth extraction	III	No
4	Female	81	Risedronate	20	Maxilla	Tooth extraction	III	No
5	Female	80	Alendronate	5	Maxilla	Spontaneous	III	No
6	Female	77	Risedronate	4	Mandible	Tooth extraction	III	No
7	Female	67	Risedronate	5	Maxilla	Tooth extraction	III	No
8	Female	82	Risedronate	10	Mandible	Tooth extraction	III	No
9	Female	76	Risedronate	20	Mandible	Tooth extraction	II	No
10	Female	80	Risedronate	10	Mandible	Root canal therapy	III	No

**Table 2 jcm-11-02145-t002:** The number of variant candidates in MRONJ patients.

Variant	SNPs				InDels			
	Saliva; Blood	Saliva; Lesion	Saliva; Blood; Lesion	Lesion-Specific	Saliva; Blood	Saliva; Lesion	Saliva; Blood; Lesion	Lesion-Specific
Intragenic	157	52	3199	611	38	202	238	750
Exonic	73	31	1866	18	4	41	40	46
Intronic	67	16	1082	556	31	138	180	624
Exonic; Splicing	-	-	1	-	-	1	-	-
Splicing	-	-	8	-	1	6	9	2
5’UTR	5	4	144	16	1	12	5	36
3’UTR	12	1	98	21	1	4	4	42
5’UTR;3’UTR	-	-	-	-	-	-	-	-
Intergenic	33	15	281	416	4	12	15	213
Upstream; Downstream	-	-	1	1	-	-	-	4
Upstream	2	1	24	19	-	4	1	31
Downstream	-	1	9	13	-	-	1	7
ncRNA exonic	12	7	128	15	-	11	4	27
ncRNA splicing	-	-	-	1	-	-	-	-
ncRNA intronic	12	3	71	75	4	12	6	57
ncRNA exonic; splicing	-	-	-	-	-	-	-	-
NA	-	-	163	-	-	-	-	-
Total variants	216	79	3876	1151	46	241	265	1089
(Total Genes)	142	54	2125	598	41	205	230	687

**Table 3 jcm-11-02145-t003:** Effects of variants in the exonic regions.

Variant Effects	Single Nucleotide Polymorphisms (SNPs)					Insertion/Deletions (InDels)			
	Saliva; Blood		Saliva; Lesion	Saliva; Blood; Lesion	Lesion-Specific	Saliva; Blood	Saliva; Lesion	Saliva; Blood; Lesion	Lesion-Specific
Missense	43		12	882	9	-	-	-	-
Synonymous	29	10	925	5	-	-	-	-	
Unknown	1	9	52	4	1	21	22	2	
Stop gain	-	-	8	-	1	-	-	-	
Nonframeshift insertion	-	-	-	-	-	6	3	13	
Nonframeshift deletion	-	-	-	-	-	6	5	20	
Frameshift insertion	-	-	-	-	2	6	5	5	
Frameshift deletion	-	-	-	-		3	5	6	
Total variants	73		31	1866	18	4	42	40	46
(Total Genes)	(49)		(21)	(1261)	(12)	(3)	(37)	(38)	(34)

**Table 4 jcm-11-02145-t004:** Annotation of rare variants with deleterious effects.

Type	Position	ID	REF	ALT	Gene	Nucleotide Change	Aminoacid Change	Exonic Function	SIFT	Polyphen2 HVAR/HDIV
SNPs	Chr1: 145368473	rs1043749	G	C	*NBPF9*	./.	./.	unknown	./.	./.
	Chr1: 145368518	rs61813437	C	T	*NBPF9*	./.	./.	unknown	./.	./.
	Chr11: 1213275	rs71251383	G	A	*MUC5AC*	./.	./.	unknown	./.	./.
	Chr12: 53343033	rs78514003	G	A	*KRT18*	c.G76A	p.A26T	missense SNV	0.549, T	0.003, B /0.005, B
	Chr12: 53343036	rs77825282	C	T	*KRT18*	c.C79T	p.R27W	missense SNV	0.032, D	0.001, B /0.001, B
	Chr12: 53343040	rs74379840	C	A	*KRT18*	c.C83A	p.P28Q	missense SNV	0.098, T	0.716, P /0.982, D
	Chr12: 53343051	rs74953757	G	T	*KRT18*	c.G94T	p.A32S	missense SNV	0.171, T	0.688, P /0.91, P
	Chr12: 53343059	rs78343594	C	A	*KRT18*	c.C102A	p.S34R	missense SNV	0.001, D	0.081, B /0.087, B
	Chr12: 53343069	rs77999286	G	T	*KRT18*	c.G112T	p.G38C	missense SNV	0.009, D	0.923, D /0.988, D
	Chr12: 53343084	rs75441140	G	C	*KRT18*	c.G127C	p.G43R	missense SNV	0.001, D	0.554, P /0.949, P
	Chr12: 53343099	.	G	A	*KRT18*	c.G142A	p.V48M	missense SNV	0.147, T	0.11, B /0.642, P
	Chr12: 53343105	rs78479490	C	T	*KRT18*	c.C148T	p.R50C	missense SNV	0.299, T	0.004, B /0.003, B
	Chr12: 53343124	rs76183244	G	A	*KRT18*	c.G167A	p.G56D	missense SNV	0.054, T	0.015, B /0.017, B
	Chr12: 53343148	.	C	G	*KRT18*	c.C191G	p.A64G	missense SNV	0.718, T	0.009, B /0.028, B
	Chr12: 53343158	rs77364359	A	G	*KRT18*	c.A201G	p.I67M	missense SNV	0.493, T	0.0, B /0.0, B
	Chr13: 25670877	rs112107735	G	A	*PABPC3*	c.G541A	p.A181T	missense SNV	0.042, D	0.253, B /0.627, P
	Chr13: 25670907	rs76264750	C	A	*PABPC3*	c.C571A	p.P191T	missense SNV	1.0, T	0.0, B /0.0, B
	Chr13: 25670919	rs76861216	A	G	*PABPC3*	c.A583G	p.I195V	missense SNV	1.0, T	0.023, B /0.005, B
	Chr13: 25670953	rs74040928	G	A	*PABPC3*	c.G617A	p.R206H	missense SNV	0.089, T	0.043, B /0.114, B
	Chr13: 25670955	rs79397892	C	T	*PABPC3*	c.C619T	p.L207F	missense SNV	0.192, T	0.978, D /0.999, D
	Chr13: 25670988	rs74564616	T	G	*PABPC3*	c.T652G	p.L218V	missense SNV	0.287, T	0.041, B /0.027, B
	Chr13: 25671027	rs78826513	A	G	*PABPC3*	c.A691G	p.K231E	missense SNV	0.001, D	0.953, D /0.995, D
	Chr13: 25671172	rs79593984	C	A	*PABPC3*	c.C836A	p.T279K	missense SNV	1.0, T	0.0, B /0.0, B
	Chr13: 25671214	rs201081849	T	G	*PABPC3*	c.T878G	p.V293G	missense SNV	1.0, T	0.0, B /0.0, B
Indels	Chr1: 17085999	rs59375146	CCCCG	C	*MST1L*	./.	./.	frameshift deletion	./.	./.
	Chr9: 95237024	.	CTCATCA	CTCATCATCA	*ASPN*	./.	./.	Non-frameshift deletion	./.	./.
	Chr9: 95237024	.	CTCATCA	CTCA	*ASPN*	./.	./.	Non-frameshift deletion	./.	./.
	Chr12: 7045891	rs797045323	ACAGCAGCAG CAGCAGAGCAG CAGCAG	ACAGCAGCAGCAG	*ATN1*	./.	./.	Non-frameshift insertion	./.	./.
	Chr12: 7045891	rs797045323	ACAGCAGCAG CAGCAGCAGCA GCAGCAG	ACAGCAGCAG CAGCAGCAGCA GCAGCAGCAGCAG	*ATN1*	./.	./.	Non-frameshift insertion	./.	./.
	Chr13: 25671149	rs368285293	ACGG	A	*PABPC3*	./.	./.	Non-frameshift deletion	./.	./.
	Chr13: 78272267	rs201380414	T	TGG	*SLAIN1*	./.	./.	frameshift insertion	./.	./.

## Data Availability

The WES data generated in this study are available from the corresponding author upon request.

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
