# Peer review of "Identification of Potentially Pathogenic Variants Associated with Recurrence in Medication-Related Osteonecrosis of the Jaw (MRONJ) Patients Using Whole-Exome Sequencing"

_jcm, 2022, doi:10.3390/jcm11082145_

Round 1

Reviewer 1 Report

Appreciate the study and it is well written.

The minor grammatical errors have been highlighted in the manuscript.

Reviewer 2 Report

Manuscript ID: jcm-1661119

Title: Identification of potentially pathogenic variants associated with recurrence in Medication-Related OsteoNecrosis of the Jaw (MRONJ) patients using whole-exome sequencing

1.What is the main question addressed by the research?

To assess genetic variants associated with medication-related osteonecrosis of the jaw (MRONJ), using whole-exome sequencing.

2.Is it relevant and interesting?

The article is relevant and interesting.

3.How original is the topic?

The topic is current.

4.What does it add to the subject area compared with other published material?

The authors have collected and analyzed a great deal of recent data.

5.Is the paper well written?

Yes, the article is well written.

6.Is the text clear and easy to read?

Yes, but moderate English editing is required.

7.Are the conclusions consistent with the evidence and arguments presented?

Yes, the conclusions consistent with the evidence and arguments presented but further studies are necessary to confirm authors’ hypothesis.

8.Do they address the main question posed?

Yes, the Authors addressed the main question posed.

Other comments:

  • English language: Moderate English editing is required.
  • Abstract: To attract the reader's attention, please clarify the target of the article, and structure the abstract.
  • Introduction: This section needs some improvements. I recommend reading the Meeting Report of the first Workshop of European task force on MRONJ: article by Schiodt et al [PMID: 31325201]. Please refer to wide range of medications classified as tyrosine kinase inhibitors, monoclonal antibodies (including tocilizumab), mammalian target of rapamycin inhibitors, radiopharmaceuticals, selective estrogen receptor modulators, and immunosuppressants implicated in osteonecrosis of the jaws. I recommend Chang et al's article entitled "Current Understanding of the Pathophysiology of Osteonecrosis of the Jaw" to implement this section [PMID: 30155844]. Please refer to relationship between oncological symptoms (like numb chin syndrome) and MRONJ [PMID: 29680775]. Please refer to efficacy of drug holiday in MRONJ management [PMID: 32568416].
  • Materials and Methods: This section was properly prepared
  • Results: This section was properly prepared
  • Discussion: What is the main theme that emerges from the authors' analysis? Is the study design a limitation? Please improve.
  • Conclusion: This section was properly prepared but further studies are necessary to confirm authors’ hypothesis.

After making the indicated changes, I am available for a second round of peer review.

Thanks for the opportunity to review this manuscript.

Reviewer 3 Report

The manuscript entitled “Identification of Potentially Pathogenic Variants Associated with Recurrence in Medication-Related Osteonecrosis of the Jaw (MRONJ) Patients using Whole-Exome Sequencing ” submitted to JCM aims to try to identify genetic variants associated with MRONJ, using whole-exome sequencing (WES).

The manuscript appears interesting with innovative insights into the pathogenetic mechanisms of this disease.

I have some suggestions to improve deeply the quality of the manuscript, enriching the text with further notions.

  • English language: Minor spell check is required.
  • Abstract: Please, revise the text as in some parts it seems redundant and the English form is not always correct.
  • Introduction: This section needs some improvements.
    I recommend to report to the last AAOMS update [DOI:https://doi.org/10.1016/j.joms.2022.02.008].
  • “For patients suffering from advanced stage (stage III) 51 MRONJ, invasive surgery, including jaw resection, can yield optimal mucosal healing” I suggest to refer to early stage management too [DOI: 10.1016/j.joms.2020.05.037 - DOI: 10.1016/j.jcms.2018.12.014].
  • “The incidence of MRONJ is relatively lower in the oral bisphosphonate group for 57 osteoporosis than in the intravenous bisphosphonate group for cancer. Kim et al. (2021) 58 reported that 21 cases of BRONJ per 100,000 person-years occurred in Korean osteoporosis 59 patients after exposure to medication for more than 4 years [3]. In 2008, researchers stud- 60 ied the prevalence of BRONJ in 600,000 patients taking bisphosphonates. Of them, 0.04% 61 (1 in 2,300) of patients were estimated to have BRONJ, and the average age of the patients 62 was 70 years (33-88 years old) in Korea [4, 5].”
    In my opinion this section could be reduced or deleted. This is not a manuscript about the incidence on MRONJ patients.
    I suggest the author to focus the attention on MRONJ onset theory including several recent reports [doi: 10.3390/jcm10204762 - DOI: 10.1177/0963689720948497 – doi: 10.1186/s41232-018-0074-9 -  DOI: 10.3390/ph13120423]
  • Methods: This section is well prepared
  • Results: Please, do not forget to explain in a better way the graphics.
  • Discussion: A study limitations section must to be included and well analyzed.
  • Conclusion: This section was properly prepared but further studies are necessary to confirm authors’ hypothesis.
  • Please, add an Abbreviation part at the end of the manuscript

After the suggested improvements, I think it is necessary to re-evaluate the text.

Round 2

Reviewer 2 Report

After the changes made, the article may be suitable for publication.

Reviewer 3 Report

The manuscript still needs to be implemented;
I suggest referring to recent literature to improve the text from a scientific point of view.
As previously suggested there are several points to be improved by referring to recent studies on onj pathology.
